

# Beyond the carapace: skull shape variation and morphological systematics of long-nosed armadillos (genus *Dasypus*)

Lionel Hautier[1,2], Guillaume Billet[3], Benoit de Thoisy[4,5] and Frédéric Delsuc[1]

[1] Institut des Sciences de l'Evolution, UMR5554, CNRS, IRD, EPHE, Université de Montpellier, Montpellier, France
[2] Mammal Section, Life Sciences, Vertebrate Division, The Natural History Museum, London, UK
[3] Sorbonne Universités, CR2P, UMR 7207, CNRS, Université Paris 06, Museum national d'Histoire naturelle, Paris, France
[4] Institut Pasteur de la Guyane, Cayenne, French Guiana, France
[5] Kwata NGO, Cayenne, French Guiana, France

## ABSTRACT

**Background:** The systematics of long-nosed armadillos (genus *Dasypus*) has been mainly based on a handful of external morphological characters and classical measurements. Here, we studied the pattern of morphological variation in the skull of long-nosed armadillos species, with a focus on the systematics of the widely distributed nine-banded armadillo (*Dasypus novemcinctus*).

**Methods:** We present the first exhaustive 3D comparison of the skull morphology within the genus *Dasypus*, based on micro-computed tomography. We used geometric morphometric approaches to explore the patterns of the intra- and interspecific morphological variation of the skull with regard to several factors such as taxonomy, geography, allometry, and sexual dimorphism.

**Results:** We show that the shape and size of the skull vary greatly among *Dasypus* species, with *Dasypus pilosus* representing a clear outlier compared to other long-nosed armadillos. The study of the cranial intraspecific variation in *Dasypus novemcinctus* evidences clear links to the geographic distribution and argues in favor of a revision of past taxonomic delimitations. Our detailed morphometric comparisons detected previously overlooked morphotypes of nine-banded armadillos, especially a very distinctive unit restricted to the Guiana Shield.

**Discussion:** As our results are congruent with recent molecular data and analyses of the structure of paranasal sinuses, we propose that *Dasypus novemcinctus* should be regarded either as a polytypic species (with three to four subspecies) or as a complex of several distinct species.

Corresponding author
Lionel Hautier,
lionel.hautier@umontpellier.fr

# INTRODUCTION

With their Pan-American distribution, long-nosed armadillos (genus *Dasypus*) constitute an understudied model for Neotropical biogeography. They are the most taxonomically diverse and widespread extant xenarthrans. The genus *Dasypus* traditionally comprises

seven extant species (*Dasypus novemcinctus*, *Dasypus hybridus*, *Dasypus septemcinctus*, *Dasypus kappleri*, *Dasypus pilosus*, *Dasypus mazzai*, and *Dasypus sabanicola*; *Wetzel, 1985*; *Wilson & Reeder, 2005*; *Feijo & Cordeiro-Estrela, 2014*) and two extinct ones (*Dasypus bellus* and *Dasypus punctatus*; *Castro et al., 2013a, 2013b*; *Castro, 2015*). In spite of being one of the earliest diverging cingulate lineages (*Gaudin & Wible, 2006*; *Delsuc et al., 2012*; *Gibb et al., 2016*), the early evolutionary history of dasypodines remains poorly known (*Castro, 2015*). Only three extinct genera are currently recognized among the Dasypodini: *Anadasypus* from the middle Miocene of Colombia and late Miocene of Ecuador (*Carlini, Vizcaíno & Scillato-yané, 1997*; *Carlini et al., 2013*), *Pliodasypus* from the late Pliocene of Venezuela (*Castro et al., 2014*), and *Propraopus* from the middle Pleistocene–early Holocene of South America (*Castro et al., 2013a*).

Aside from the widespread nine-banded armadillo (*Dasypus novemcinctus*), all extant long-nosed armadillos are restricted to South America. Some species are sympatric in certain areas, resulting in competition and possibly supporting divergent behaviors and morphologies. The nine-banded armadillo is likely to be the most abundant armadillo in tropical forests (*Wetzel & Mondolfi, 1979*; *Loughry & McDonough, 1998*), and it has the widest distribution of all extant xenarthran species. Its distribution is thought to cover much of South and Central America and parts of North America, and it ranges from the southeastern United States to northwestern Argentina and Uruguay (*McBee & Baker, 1982*; *Loughry & McDonough, 2013*). The species ability to disperse quickly, as well as its opportunistic and generalist mode of life, could partly explain this broad distribution (*Smith & Doughty, 1984*; *Loughry & McDonough, 1998, 2013*) as well as its rapid historical expansion into the United States (*Taulman & Robbins, 2014*). Such a wide geographical distribution, combined with long-recognized morphological variations (*Peters, 1864*; *Gray, 1873*; *Allen, 1911*; *Lönnberg, 1913*; *Russell, 1953*), raise the possibility that major taxonomic subgroups have been overlooked, be it at the subspecific or even specific level.

As its vernacular name implies, the genus *Dasypus* is characterized by a long, slender rostrum, which represents at least 55% of the length of the head (*Gardner, 2008*). The different species are usually distinguished by body and cranial measurements, color differences, and morphological features of the carapace, such as the number of movable bands and scutes across the body and the number and shape of osteodermal foramina (*Wetzel, 1985*; *Feijo & Cordeiro-Estrela, 2016*). The carapace is a hallmark of armadillos, and constitutes such a unique feature for mammals that it has dominated the attention of early and modern anatomists and, as a result, partly jeopardized the classification of the group. Its morphology, chiefly the number of movable bands, has been extensively used in systematic studies. However, even in the so-called nine-banded armadillo, the number of movable bands can vary from seven to 11 (*Wetzel & Mondolfi, 1979*; *McBee & Baker, 1982*). Yet early on, in his Systema Naturae, *Linnaeus (1758*: 51*)* cast doubt on the usefulness of the number of movable bands as a criterion to distinguish *Dasypus* species (in this case *Dasypus septemcinctus* from *Dasypus novemcinctus*). Since then, a number of authors have raised the question whether such external features could be confidently used for systematic purposes. *Wetzel & Mondolfi (1979*: 46*)* argued that "although many scientific names of armadillos are based on the number of movable bands, it is proposed here that

for vernacular names we discontinue using this variable characteristic and base names upon unique or more consistent features." In the early 20th century, *Hamlett (1939)* made similar observations on the nine-banded armadillo. He considered it impossible to recognize external variations at a subspecific level, and concluded that "cranial characters appear to offer the only promise for subspecific analysis of the species" (*Hamlett, 1939*: 335). We decided to further investigate Hamlett's idea, since no large review of the dasypodine cranial variation has been undertaken to date.

This study aims to elucidate the pattern of morphological variation seen in the skull of long-nosed armadillos, with a focus on the nine-banded armadillo. Geometric morphometric data were collected for most *Dasypus* species using micro-computed tomography ($\mu$CT). The main questions asked in the present study are whether different patterns of variation in skull shape can be characterized among and within long-nosed armadillo species, and if those patterns could be linked to factors such as taxonomy, geographical distribution, skull size, or sexual dimorphism. Our ultimate goals are to reconstruct the details of the biogeographic distribution of the widespread nine-banded armadillo at the continental scale and to lay the path for a new integrative taxonomy of long-nosed armadillos. A greater understanding of the morphological diversity and patterns of evolution for long-nosed armadillos is timely to effectively conserve these species, and will also serve to deepen our knowledge of their peculiar evolution and biology (*Loughry & McDonough, 2013*).

# MATERIALS AND METHODS

## Biological samples

The material studied came from the collections of the Muséum national d'Histoire naturelle (MNHN, collections Zoologie et Anatomie comparée, Mammifères et Oiseaux) in Paris (France), the Natural History Museum (BMNH) in London (UK), the Naturalis Biodiversity Center (NBC) in Leiden (Holland), the Royal Ontario Museum (ROM) in Toronto (Canada), the Museum of Natural Science of the Louisiana State University (LSU) in Baton Rouge (USA), the American Museum of Natural History (AMNH) in New York (USA), the National Museum of Natural History (NMNH) in Washington (USA), the Instituto de Pesquisas Científicas e Tecnológicas do Estado do Amapá (IEPA) in Macapá (Brazil), and the Muséum d'Histoire Naturelle in Geneva (Switzerland), the KWATA association in Cayenne (French Guiana), and the personal collection of Pierre Charles-Dominique in Montpellier (France). We analyzed 128 skulls belonging to five *Dasypus* species (see Table S1 for a complete list of specimens): *Dasypus novemcinctus*, *Dasypus hybridus*, *Dasypus septemcinctus*, *Dasypus kappleri*, and *Dasypus pilosus* (no data was available for *Dasypus mazzai* and *Dasypus sabanicola*). With these data we performed a preliminary assessment of the average amounts of cranial variation at the specific level among different populations of *Dasypus novemcinctus* from French Guiana, Guyana, Suriname, Ecuador, Brazil, Venezuela, Colombia, Costa Rica, Belize, Bolivia, Argentina, Paraguay, Uruguay, Panama, Nicaragua, Honduras, Guatemala, Mexico, Peru, and the USA (Table S1). Juvenile, subadult, and adult specimens were considered in order to take into account the effect of age, size, and differential growth on the dataset. Several studies
(*Hensel, 1872*; *Russell, 1953*; *Ciancio et al., 2012*) showed that long-nosed armadillos possess tooth replacement, as is typical for mammals, and that the eruption of permanent teeth occurs relatively late, as observed in afrotherians (*Asher & Lehmann, 2008*). Accordingly, we used eruption of the teeth, suture closure, and size (i.e., skull length) as criteria to identify adult specimens in our dataset. Specimens were considered as juveniles when they displayed milk teeth and an open suture between the basioccipital and the basisphenoid. Subadults were characterized by erupting permanent teeth. Adult specimens always showed a closed basioccipital/basisphenoid suture and a fully erupted permanent dentition. Except when specified, only adults and subadults specimens were considered in the morphometrical analyses.

## Geometric morphometric methods

Due to the limitations of the classical qualitative descriptive approach, geometric morphometrics represents a good complementary technique by which to examine intraspecific shape variation. Digital data for all specimens were acquired using X-ray μCT at the University of Montpellier (France), at the Natural History Museum (London, UK), and at the AST-RX platform MNHN (Paris, France). Three-dimensional reconstruction and visualization of the skulls were performed using stacks of digital μCT images with AVIZO v. 6.1.1 software (Visualization Sciences Group, Burlington, MA, USA). The mandibles and crania of armadillos were quantified with 10 and 82 anatomical landmarks respectively (Fig. 1; Tables 1 and 2) using ISE-MeshTools (version 1.3.1; *Lebrun, 2014*). These landmarks were inspired by previous studies performed on different mammalian taxa (*Hautier, Lebrun & Cox, 2012*; *Hautier et al., 2014*). Considering the tendency toward the reduction of the number of teeth in *Dasypus* (*Allen, 1911*), specimens often lack the last dental locus, which corresponds to a molar (*Ciancio et al., 2012*). In order to avoid producing an artificial shortening of the entire tooth row, we decided not to place a landmark at the end of the tooth row, as it is commonly the case, but after the last premolar locus (in both mandible and maxilla). Since skulls were often incomplete, the number of landmarks was adjusted to incorporate the maximal morphological variation in a maximum number of individuals. This number differed when we performed analyses including all *Dasypus* species (10 and 70 landmarks for the mandible and the cranium respectively) or only *Dasypus novemcinctus* (10 and 82 landmarks for the mandible and the cranium respectively).

All configurations (sets of landmarks) were superimposed using the Procrustes method of generalized leastsquares superimposition (GLS scaled, translated, and rotated configurations so that the intralandmark distances were minimized) following the methods of *Rohlf (1999)* and *Bookstein (1991)*. Subsequently, mandibular and cranial forms of each specimen were represented by centroid size S, and by multidimensional shape vector v in linearized Procrustes shape space. Shape variability of the skull and the mandible was visualized by principal component analysis (PCA) of shape (*Dryden & Mardia, 1998*). Analysis and visualization of patterns of shape variation were performed with the interactive software package MORPHOTOOLS (*Specht, 2007*; *Specht, Lebrun & Zollikofer, 2007*; *Lebrun, 2008*; *Lebrun et al., 2010*).

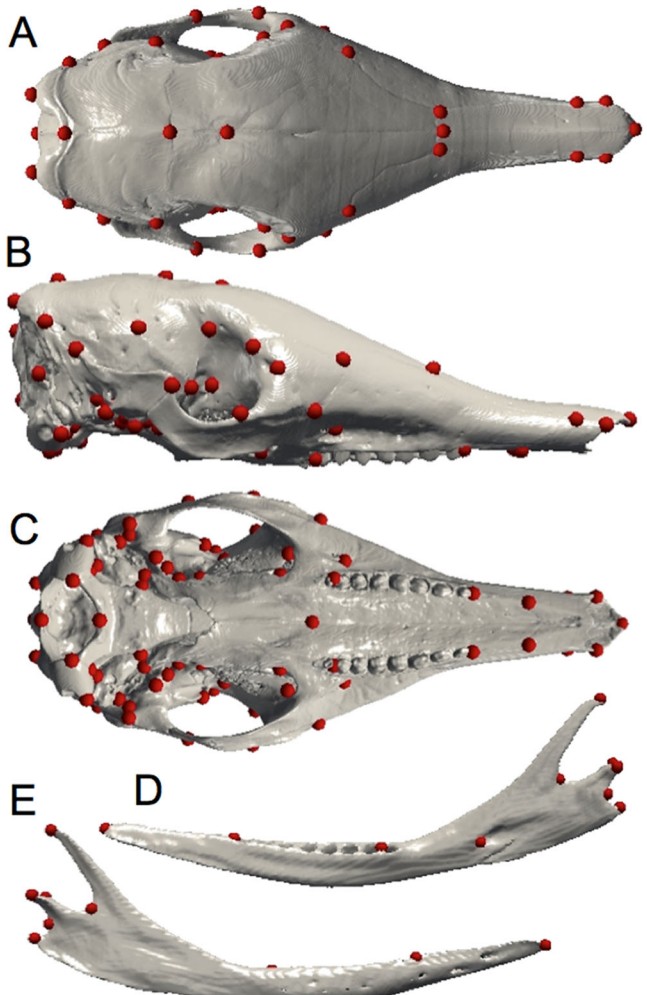

**Figure 1 Landmarks digitized on the mandible and the skull.** Dorsal (A), lateral (B), and ventral views of the cranium; medial (C) and lateral (D) views of the mandible.

To account for the potentially confounding effects of size allometry on shape of the cranium and mandible at the intraspecific level (i.e., within *Dasypus novemcinctus*), size-corrected shapes were obtained as follows: we conducted a multivariate regression of Procrustes residuals on the logarithm of the centroid size, yielding an allometric shape vector (ASV). The ASV represents a direction in shape space, which characterizes mandibular and cranial allometric patterns. All crania and mandibles of *Dasypus novemcinctus* were then projected on ASVs, the residuals representing the component of skull shape that is independent of size. Shape variation independent of size was visualized by PCA of shape using MORPHOTOOLS. PCres corresponds to principal components of a PCA performed on shape data corrected for allometry. The same analyses could not be performed at the interspecific level due to large differences in sample size.

Multivariate analyses of variance (MANOVA) were performed on the principal component scores of mandibular and cranial mean shapes in order to assess the effects of

**Table 1 Definitions of the landmarks used on the mandible.**

| Numbers | Definition |
|---------|-----------|
| 1 | Most anterior point of the mandible |
| 2 | Most anterior point of the alveolar margin of the tooth row |
| 3 | Most posterior point of the premolar tooth row |
| 4 | Tip of the coronoid process |
| 5 | Point at the maximum of concavity between the coronoid and the condyloid processes |
| 6 | Most lateral point of the articular surface of the condyle |
| 7 | Most medial point of the articular surface of the condyle |
| 8 | Point at the maximum of concavity between the condyloid and the angular |
| 9 | Tip of the angular process |
| 10 | Mandibular foramen |

**Table 2 Definitions of the landmarks used on the cranium.**

| Numbers | Definition |
|---------|-----------|
| 1 | Most anterodorsal point of the nasal suture |
| 2 | Intersection between internasal and interfrontal sutures |
| 3 | Intersection between interparietal and interfrontal sutures |
| 4 | Intersection between interparietal and supraoccipital |
| 5 | Most distal point of the supraoccipital |
| 6 and 7 | Intersection between frontal, maxillar, and nasal sutures |
| 8 and 9 | Most dorsomedial point of the orbit (i.e., minimal interorbital length) |
| 10 and 11 | Most posterolateral point of the supraoccipital |
| 12 and 28 | Most anterolateral point of the premaxillar/nasal suture |
| 13 and 29 | Intersection between premaxillar, maxillar, and nasal sutures |
| 14 and 30 | Intersection between the lacrimal, maxillar, and frontal sutures |
| 15 and 31 | Anteroventral margin of the lacrimal foramen |
| 16 and 32 | Anteroventral margin of the upper ethmoid foramen |
| 17 and 33 | Most anterior point of the squamosal, frontal, and alisphenoid sutures |
| 18 and 34 | Most dorsal point of the maxillary foramen |
| 19 and 35 | Most dorsal point of the infraorbital foramen |
| 20 and 36 | Most anteroventral point of the sphenopalatine fissure |
| 21 and 37 | Most dorsal point of the jugal/maxillar suture |
| 22 and 38 | Most dorsal point of the jugal/squamosal suture |
| 23 and 39 | Most posterior point of the postglenoid process |
| 24 and 40* | Most posterodorsal point of the zygomatic part of the squamosal |
| 25 and 41 | Intersection between the frontal, squamosal, and parietal sutures |
| 26 and 42* | Most dorsal point of sulcus for external acoustic meatus on squamosal |
| 27 and 43 | Intersection between the parietal, squamosal, and supraoccipital sutures |
| 44 and 60 | Most posterior point of the premaxillar/maxillar suture in ventral view |
| 45 and 61 | Most anterior point of the alveolar margin of the premolar tooth row |
| 46 and 62 | Most posterior point of the alveolus margin of the premolar tooth row. |
| 47 and 63 | Intersection between the lacrimal/maxillar suture and the zygomasseteric crest in ventral view |

| Numbers | Definition |
|---|---|
| 48 | Intersection between maxillar and palatine sutures |
| 49 and 64 | Most posterolateral point of the pterygoid wings |
| 50 and 65 | Transverse canal foramen |
| 51 and 66 | Most anterodorsal point of the *foramen ovale* |
| 52 and 67 | Most ventral of the alisphenoid/squamosal suture |
| 53 and 68 | Most lateral point between the basioccipital/basisphenoid sutures |
| 54* and 69* | Most posterolateral point of the jugular foramen |
| 55* and 70* | Most posterolateral point of the hypoglossal foramen |
| 56* and 71* | Most anterolateral point of the occipital condyle |
| 57* and 72* | Intersection between the basioccipital, the occipital condyle, and the *foramen magnum* |
| 58* | Most anteroventral point of the *foramen magnum* |
| 59* | Most posterodorsal point of the *foramen magnum* |
| 73 and 74 | Intersection between the supraoccipital, exoccipital, and petrosal sutures |
| 75 and 76 | Most posterior point of the postglenoid foramen |
| 77 and 78 | Caudal palatine foramen |
| 79 and 80 | Intersection between the lacrimal/frontal suture and the orbit |
| 81 | Most posterior point of the frontal sinuses in the midline |
| 82 | Ventral tip of the tentorial process |

**Note:**
Landmarks indicated with a star were not used in the intraspecific comparisons.

different factors on mandibular and cranial shape variation: clades (species), sex, and geographic distribution (countries). Because our dataset consisted of a relatively large number of variables, PCs accounting for 90% of the interspecific shape variance in the sample were retained for analysis. MANOVAs were performed with Past 2.06 (*Hammer, Harper & Ryan, 2001*). Linear discriminant analyses (LDA) of shape coordinates were performed on the same number of PCs to assess a potential discrimination of skull morphology in relation to phylogeny (i.e., species) and geography (i.e., countries). If lumping specimens by country has little biological relevance, we consider that using such artificial boundaries also constitutes a mean to avoid making a priori hypotheses. In all cases, all LDA results were double-checked with post hoc classification methods and PCA results. When a group included only one individual, this specimen was integrated into the dataset as an ungrouped case. A skull from Panama (NMNH 171052) was not complete enough to be considered in these analyses. We then decided to perform similar analyses with a reduced set of landmarks (81 landmarks on the cranium) to enable its morphological comparison with other specimens.

## Linear measurements

Several linear measurements of the skull of *Dasypus* were calculated directly on 3D coordinates of landmarks (Fig. 2). These measurements were used to compare our results with traditional methods of species delineation.

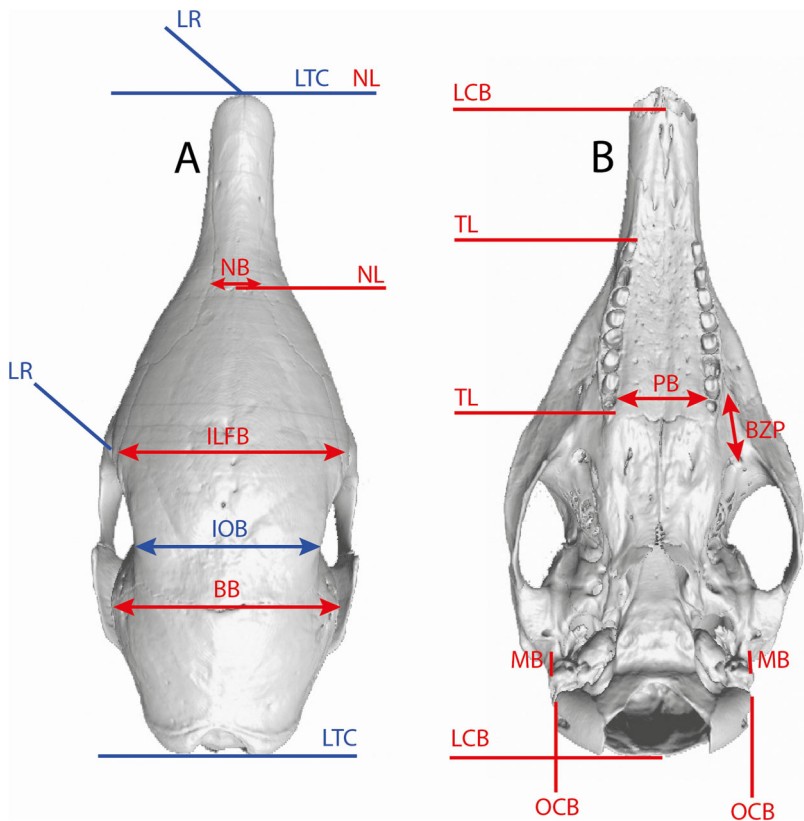

**Figure 2 Illustration of the skull linear measurements.** In blue, traditional measurements used in *Wetzel (1985)*. LTC, length between the anterior tip of the nasal and the posteriormost point of the supraoccipital; LR, rostral length; IOB, interorbital breadth; ILFB, inter lacrimal foramina breadth; BB, distance between the left and right intersections between the frontal, parietal, and squamosal sutures; NB, nasal breadth; NL, nasal length; LCB, length between the anterior tip of the premaxillar and the condyles; TL, length of the tooth row; PB, palate breadth; BZP, distance between the infraorbital and the maxillary foramina; MB, inter-meatus breadth; OCB, breadth between the lateral border of the occipital condyle.

## RESULTS

### Interspecific variation of skull shape among long-nosed armadillos

A MANOVA performed on the first 35 PCs (i.e., 90% of the variance) indicates a significant morphological differentiation of the mandibles and crania relative to species delimitations (mandible Wilks' $\lambda = 0.01164$, $F = 4.903$, $p < 0.001$; cranium Wilks' $\lambda = 0.0005897$, $F = 11$, $p < 0.001$). A multivariate regression of the shape component on size, estimated by the logarithm of centroid size, was significant for the skull (mandible Wilks' $\lambda = 0.2709$, $F = 6.637$, $p < 0.001$; cranium Wilks' $\lambda = 0.0632$, $F = 30.92$, $p < 0.001$).

Morphological differences occur among the mandibles of the five species of *Dasypus* (Fig. 3A). The first two principal components (24.73% and 16.48% of total shape variation) weakly discriminates *Dasypus pilosus* (negative values) from *Dasypus kappleri* (positive values) while all specimens of *Dasypus novemcinctus*, *Dasypus hybridus*, and *Dasypus septemcinctus* sit in the middle of the graph. These axes separate mandibles having a slender horizontal ramus, an elongated anterior part (located in front of the
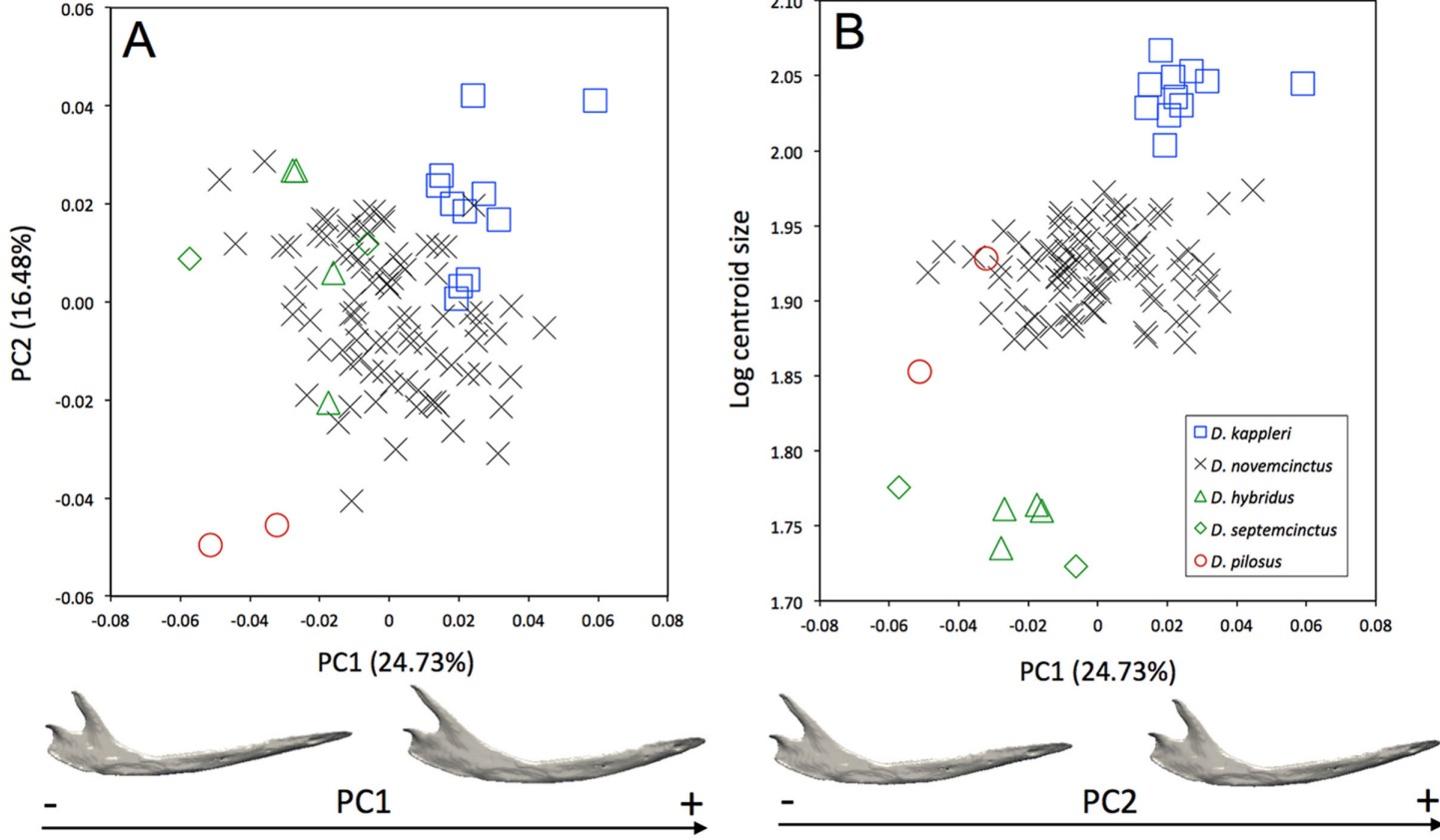

**Figure 3** **(A) Principal component analysis (PC1 vs PC2) and associated patterns of morphological transformation for the mandible of five** ***Dasypus*** **species. (B) Regression of the first principal component on the logarithm of the centroid size ($R^2 = 0.23$; $p < 0.001$).** *Symbols*: blue squares, *Dasypus kappleri*; black crosses, *Dasypus novemcinctus*; green triangles, *Dasypus hybridus*; green diamonds, *Dasypus septemcinctus*; red circles, *Dasypus pilosus*.

tooth row), a short ascending ramus, and a short, anteriorly positioned and vertically oriented coronoid process, from mandibles showing a high horizontal ramus, a short anterior region, a long ascending ramus, and an elongated and distally oriented coronoid process. Mandibles of different size are poorly discriminated along the first principal component (Fig. 3B).

The interspecific differences in the cranium of *Dasypus* are apparent in the morphospace defined by the first two principal components. Except for *Dasypus hybridus* and *Dasypus septemcinctus*, all species are well discriminated in the morphospace defined by the two first principal components (Fig. 4A), which explain 28.76% and 15.29% of the variance respectively. The first principal component unequivocally discriminates *Dasypus hybridus* and *Dasypus septemcinctus* from other species, and negatively correlates with a shortened rostrum and enlarged basicranium and braincase (Fig. 4A). *Dasypus pilosus* individuals are well discriminated on the second principal component. On PC2 (Fig. 4A), the crania of *Dasypus pilosus* appear narrower, with a long snout and smaller braincase (positive values), whereas the crania of *Dasypus novemcinctus*, *Dasypus kappleri*, *Dasypus hybridus*, and *Dasypus septemcinctus* are wider, with a shorter snout and relatively large braincase (negative values). A regression of the first principal component on the

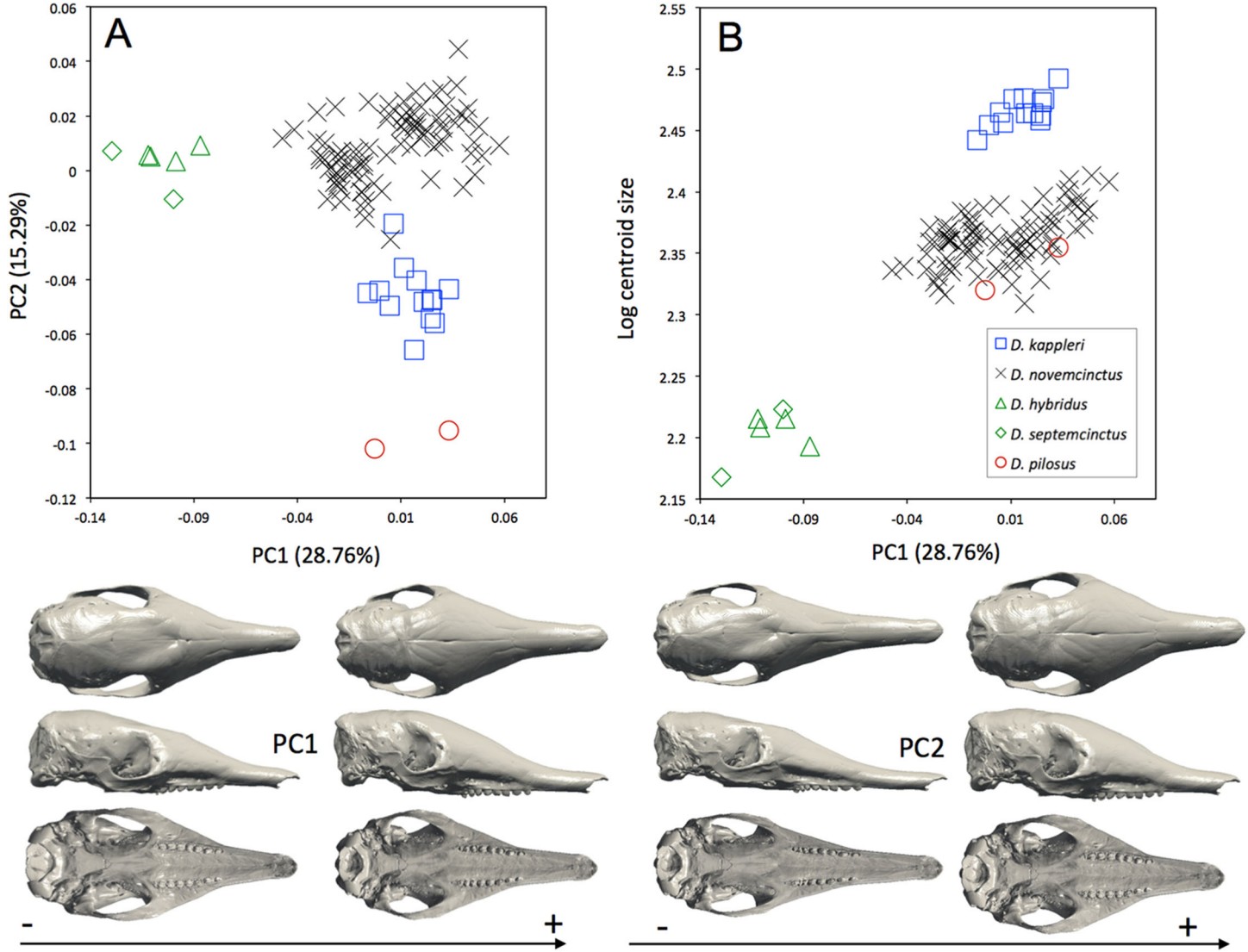

**Figure 4** **(A) Principal component analysis (PC1 vs PC2) and associated patterns of morphological transformation for crania of five *Dasypus* species. (B) Regression of the first principal component on the logarithm of the centroid size ($R^2$ = 0.50; $p$ < 0.001).** *Symbols:* blue squares, *Dasypus kappleri*; black crosses, *Dasypus novemcinctus*; green triangles, *Dasypus hybridus*; green diamonds, *Dasypus septemcinctus*; red circles, *Dasypus pilosus*.

logarithm of the centroid size (Fig. 4B) clearly shows that the five species show different size ranges. The biggest crania are long and display a longer and wider snout, whereas the smallest crania are short and wide posteriorly and characterized by a short snout. A posteriori classification methods produced 100% correct specific classification of specimens.

We also performed the same analyses, this time including juvenile specimens (Fig. S1) but excluding *Dasypus pilosus*, since it represents a clear outlier in the morphospace. All juvenile specimens of *Dasypus novemcinctus* tend to exhibit negative values for PC1, and thus appear more similar in shape to *Dasypus septemcinctus* and *Dasypus hybridus* than the adult *Dasypus novemcinctus*.

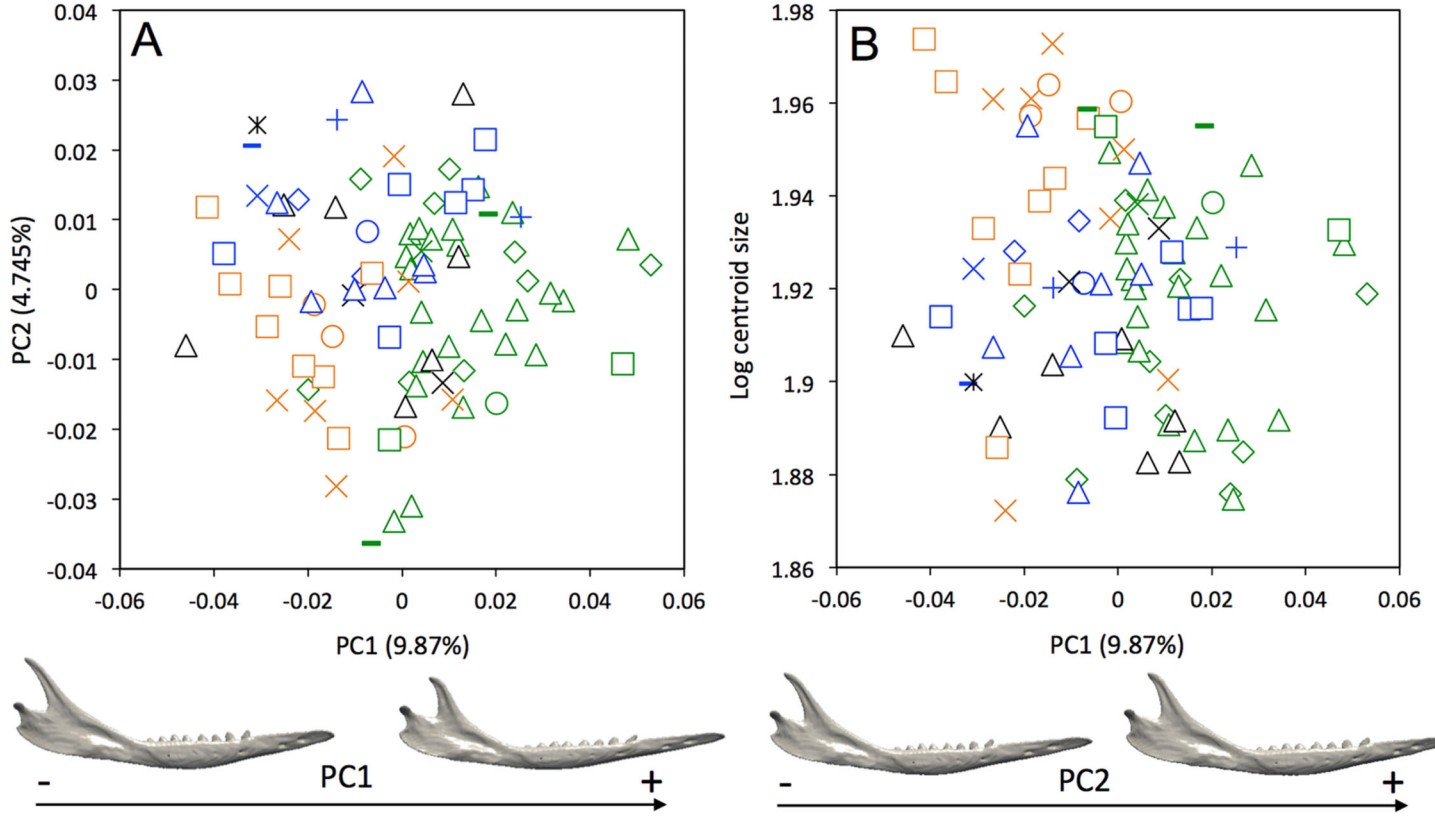

**Figure 5** **(A) Principal component analysis (PC1 vs PC2) and associated patterns of morphological transformation for mandibles of *Dasypus novemcinctus*. (B) Regression of the first principal component on the logarithm of the centroid size ($R^2 = 0.035$; $p = 0.03$).** *Symbols*: green diamonds, Bolivia; green triangle, Brazil (solid green triangles are for specimens from Amapa); green circles, Paraguay; green crosses, Peru; green squares, Uruguay; green bars, Venezuela; blue diamonds, Belize; blue "plus", Guatemala; blue bars, Honduras; blue squares, Mexico; blue crosses, Nicaragua; blue triangles, USA; blue circles, Costa Rica; black triangles, Colombia; black crosses, Ecuador; black stars, Panama; orange squares, French Guiana; orange crosses, Guyana; orange circles, Suriname.

## Intraspecific variation of skull shape in nine-banded armadillos

In specimens for which sex was available (22 females and 32 males for the mandible; 19 females and 34 males for the cranium), a MANOVA shows that there is no sexual dimorphism present in the cranial data (Wilks' $\lambda = 0.2429$, $F = 1.514$, $p = 0.182$), so sex is unlikely to be responsible for the variation observed in the cranium of *Dasypus novemcinctus*, although it might partly account for the variation in the mandible (Wilks' $\lambda = 0.2292$, $F = 2.579$, $p = 0.011$). ANOVAs also show that there is no relationship between sex and size (mandibles, $F = 0.4219$, $p = 0.52$; cranua, $F = 0.00143$, $p = 0.97$). A multivariate regression of the shape component on size was significant (mandible, Wilks' $\lambda = 0.3468$, $F = 3.454$, $p < 0.001$; cranium, Wilks' $\lambda = 0.1447$, $F = 8.446$, $p < 0.001$). When looking at the ASVs obtained with the centroid size (Fig. S2), we found that size explains 14.14% and 14.32% of the variation in the whole mandibular and cranial data sets respectively. Shape data corrected for allometry are presented in Figs. S3 and S4.

A weak intraspecific differentiation (per country) is noticeable in the mandibular morphology data (Wilks' $\lambda = 0.0001404$, $F = 1.523$, $p < 0.001$; Fig. 5A). The first principal component (9.87% of total shape variation) weakly discriminates specimens from Brazil,

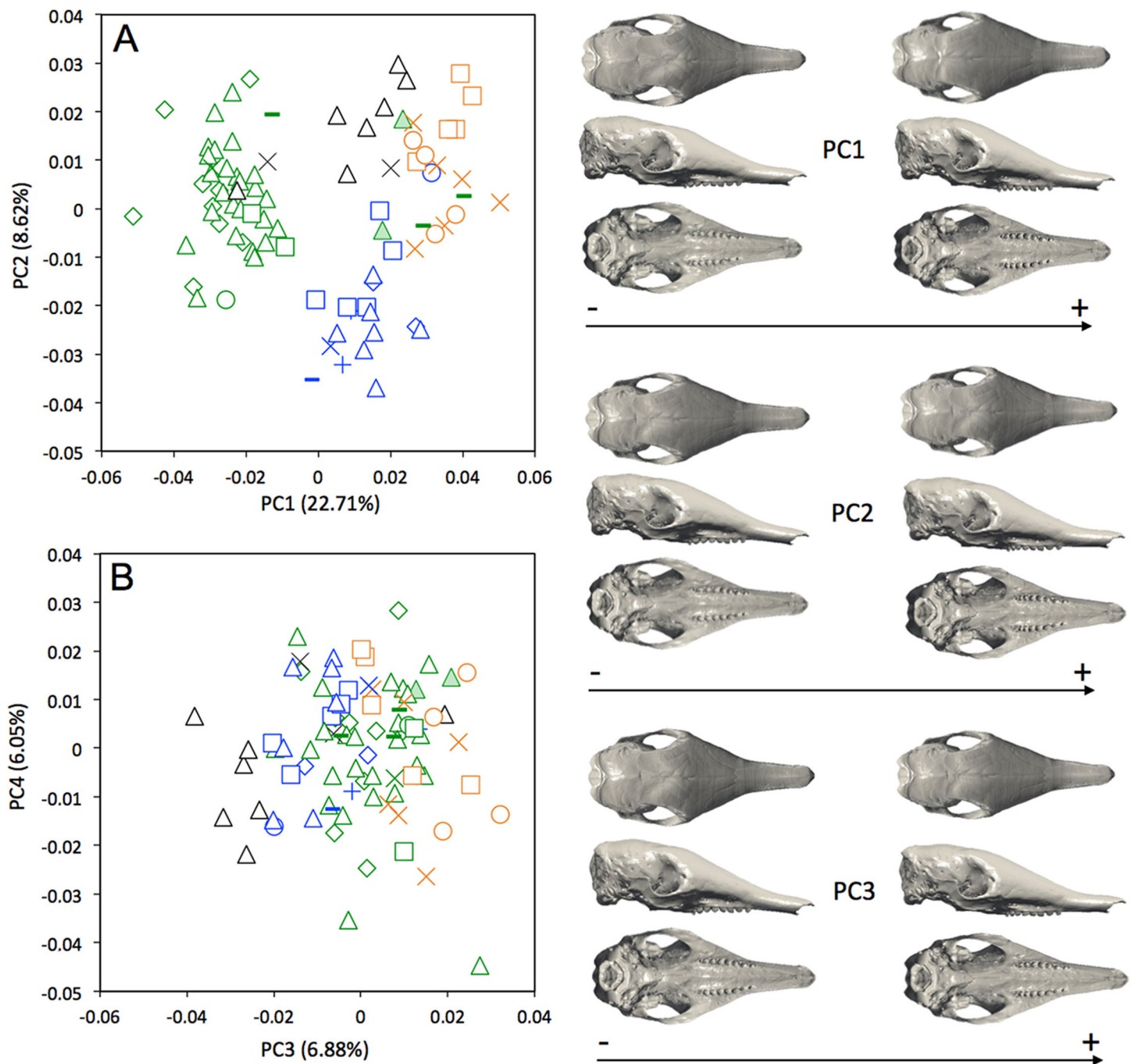

**Figure 6 Principal component analysis (A, PC1 vs PC2; B, PC3 vs PC4) and associate patterns of morphological transformation for crania of** *Dasypus novemcinctus.* *Symbols*: green diamonds, Bolivia; green triangle, Brazil (solid green triangles are for specimens from Amapa); green circles, Paraguay; green crosses, Peru; green squares, Uruguay; green bars, Venezuela; blue diamonds, Belize; blue "plus", Guatemala; blue bars, Honduras; Blue squares, Mexico; blue crosses, Nicaragua; blue triangles, USA; blue circles, Costa Rica; black triangles, Colombia; black crosses, Ecuador; orange squares, French Guiana; orange crosses, Guyana; orange circles, Suriname.

Bolivia, Paraguay, Uruguay (positive values) from other specimens (negative values). This axis separates mandibles characterized by a robust and short horizontal ramus, a long ascending ramus, a high coronoid process, a low condylar process, and a poorly

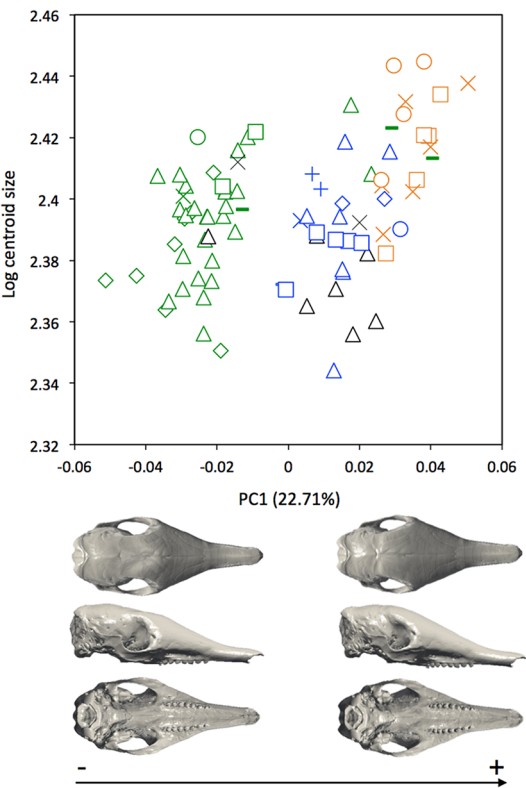

**Figure 7 Regression of the first cranial principal component (*Dasypus novemcinctus*) on the logarithm of the centroid size ($R^2 = 0.15$; $p < 0.001$).** *Symbols*: green diamonds, Bolivia; green triangle, Brazil (solid green triangles are for specimens from Amapa); green circles, Paraguay; green crosses, Peru; green squares, Uruguay; green bars, Venezuela; blue diamonds, Belize; blue "plus", Guatemala; blue bars, Honduras; blue squares, Mexico; blue crosses, Nicaragua; blue triangles, USA; blue circles, Costa Rica; black triangles, Colombia; black crosses, Ecuador; orange squares, French Guiana; orange crosses, Guyana; orange circles, Suriname.

individualized angular process, from mandibles with a slender and elongated horizontal ramus, a short ascending ramus, a low coronoid process, a high condylar process, and a well individualized angular process (Fig. 5A). In terms of shape variation, PC2 (4.745% of total shape variation) separates mandibles that show an elongated anterior part of the horizontal ramus, a short tooth row, and a high and distally oriented coronoid processes, from mandibles having a reduced anterior part of the horizontal ramus, an elongated tooth row, and a low coronoid process. We observed even less differentiation with shape data corrected for allometry (Fig. S3), which indicates that some specimens differ significantly in size. This is confirmed by a regression of the first principal component on the logarithm of the centroid size (Fig. 5B), which shows that the specimens from Brazil, Uruguay, Paraguay, Bolivia, Peru, Ecuador, Costa Rica, and Colombia are usually smaller.

A MANOVA was also used to explore if the cranial variation matches the geographical distributions of *Dasypus novemcinctus* (Wilks' $\lambda = 2.97 \times 10^{-6}$, $F = 2.157$, $p < 0.001$). When looking at the cranial morphological variation according to geographic origin (i.e., countries) (Fig. 6), several trends can be observed. PC1, accounting for 22.7% of the

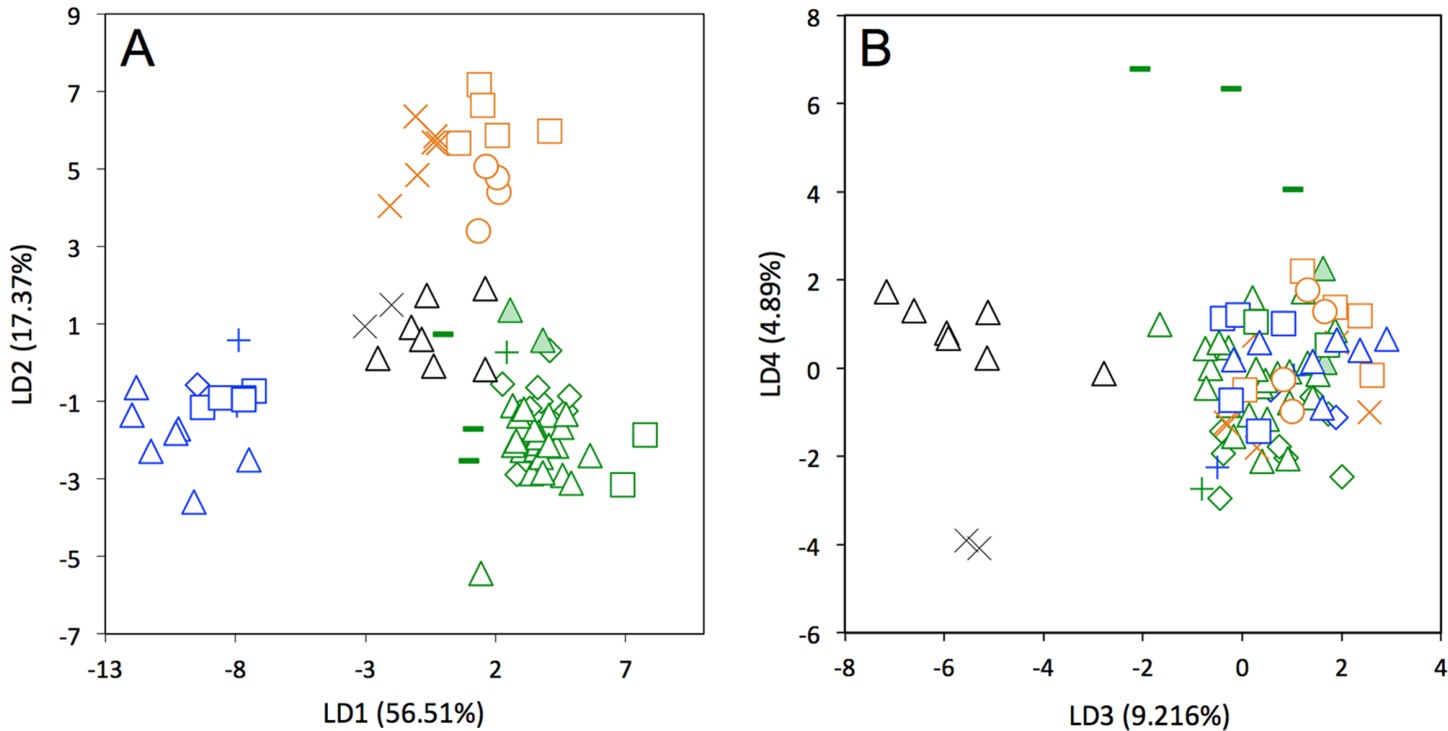

**Figure 8** Linear Discriminant Analysis (LDA) performed on cranial shape coordinates of *Dasypus novemcinctus* (A, LD1 vs LD2; B, LD3 vs LD4). *Symbols*: green diamonds, Bolivia; green triangle, Brazil (solid green triangles are for specimens from Amapa); green circles, Paraguay; green crosses, Peru; green squares, Uruguay; green bars, Venezuela; blue diamonds, Belize; blue "plus", Guatemala; blue bars, Honduras; Blue squares, Mexico; blue crosses, Nicaragua; blue triangles, USA; black triangles, Colombia; black circles, Costa Rica; black crosses, Ecuador; orange squares, French Guiana; orange crosses, Guyana; orange circles, Suriname.

overall variation, demonstrates a change in how domed the dorsal surface of the skull is, and positively correlates with an increase in snout length, a decrease in braincase size, jugals that are more extended dorsoventrally, and shorter pterygoid processes (Fig. 6A). Specific clusters are recognizable on the first principal component, with specimens from Brazil, Paraguay, Venezuela (NMNH 406700 from Clarines area, North), Ecuador (BMNH-14-4-25-86 from Gualaquiza, South East), Colombia (AMNH 136252 from Villavicencio area, Centre), Peru, Bolivia, Paraguay, and Uruguay that congregate in the negative values, whereas all other specimens lay in the positive values. PC2 is responsible for 8.6% of the variance, and describes variation in the size of the posterior part of the rostrum; it also displays variation in length of the posterior part of the palate with an anterior border of the palatine that is well behind the posterior end of the tooth row in skulls showing positive values. This axis mainly separates specimens from USA, Mexico, Belize, Honduras, Guatemala, and Nicaragua (negative values) from other specimens (positive values). We observed less specific differentiation with shape data corrected for allometry (Fig. S4), which shows that the different geographical subgroups differ significantly in size. This is confirmed by a regression of the first principal component to the logarithm of the centroid size (Fig. 7).

A LDA of shape coordinates was performed in order to take into account the entirety of morphological variation (i.e., 35 first PCs that represent 90.6% of the variance) and to

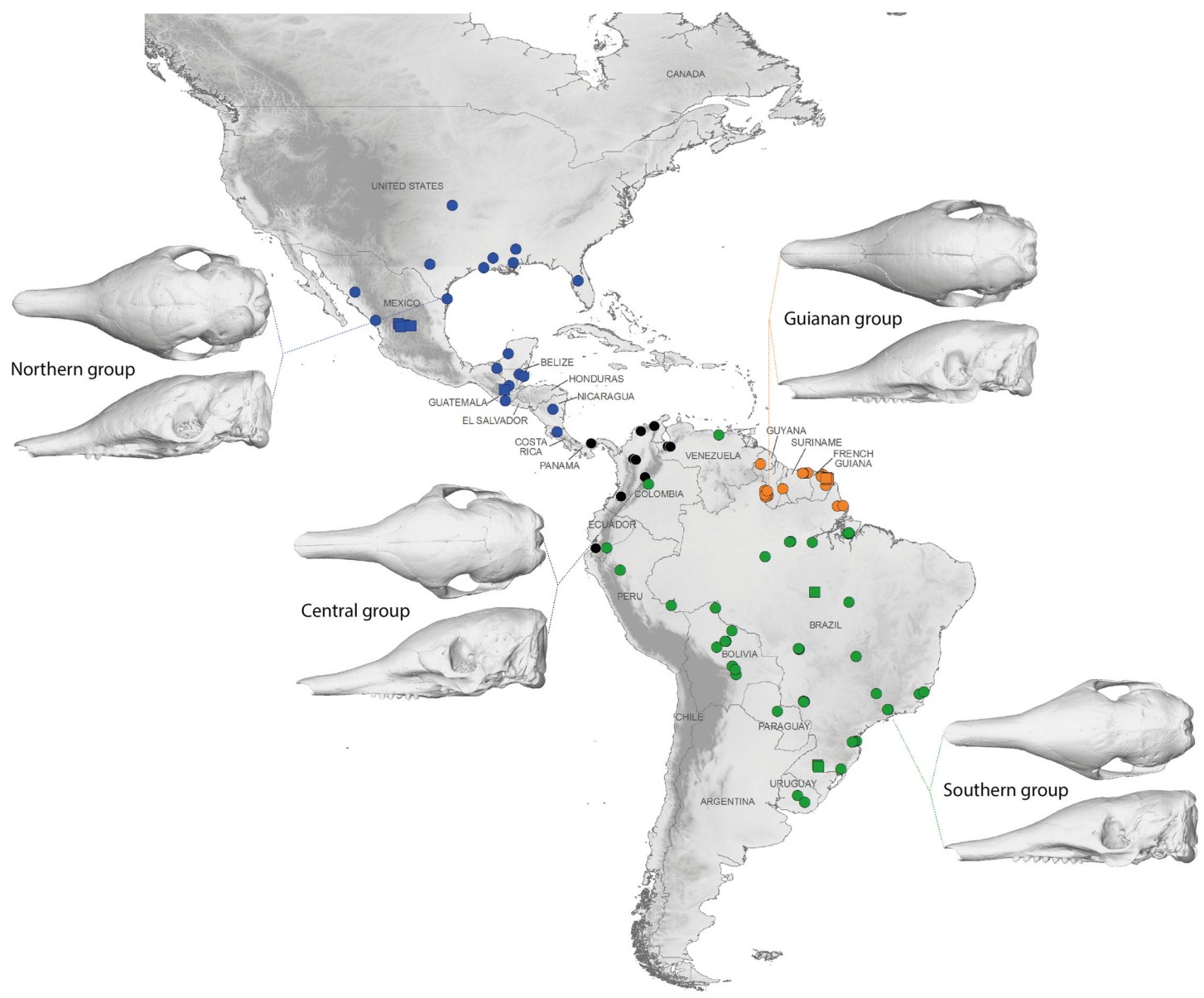

**Figure 9 Summary map showing the geographical distribution of nine-banded armadillo specimens investigated in this study and their attribution to one of the four main morphotypes defined in this study: black, Central group; blue, Northern group; green, Southern group; orange, Guianan group.** Specimens lacking precise geographical information (other than country of origin) are indicated with a square.

maximize discrimination among specimens belonging to different countries. Only countries for which we had several specimens could be considered here. Three main regional groups were clearly recovered by the analysis (Fig. 8A): a Northern morphotype, a Southern morphotype, and a group restricted to the Guiana Shield (GS). The first group from North and Central America includes specimens from the USA, Mexico, Guatemala, and Belize. The South American group gathers specimens from Brazil, Uruguay, Bolivia, Peru, Colombia, and Venezuela. Finally, specimens from French Guiana, Suriname, and Guyana congregate in a last distinctive group. Some specimens from Colombia, Venezuela, and Ecuador do not gather with any of those groups and sit in the middle of

the graph defined by the first two discriminant axes; these specimens are, however, well discriminated on the third and fourth discriminant axes (Fig. 8B) and might constitute a fourth individualized regional group among *Dasypus novemcinctus*, called hereafter the Central morphotype.

The discriminant model used to separate the regional groups, based on artificial boundaries (i.e. borders of countries), was checked using a classification phase. The same procedure was then used on under-sampled countries (i.e., when $n = 1$) to assess their affiliation to one of the four abovementioned groups. This analysis showed 95% correct classification of specimens (Table S2). Most regional misclassifications were with specimens coming from the limit of the distribution range of the groups. Two Brazilian specimens from Amapa are put together with the Guianan specimens (Table S2) and confirm previous results from the PCA, where these two specimens clearly depart from the rest of the Brazilian specimens (Fig. 6A). Three specimens from Venezuela (NMNH 406700 from Clarines area, North), Ecuador (BMNH-14-4-25-86 from Gualaquiza, South East), and Colombia (AMNH 136252 from Villavicencio area, Centre) were a posteriori classified as close to the Southern morphotype. All these specimens were collected East of the Andes (Fig. 9) and grouped with Brazilian specimens in the PCA analyses. Concerning the countries for which only one specimen was available, the classification analyses gave congruent results with the grouping proposed by the PCA: specimens from Paraguay and Peru were classified as being part of the Southern morphotype, whereas specimens from Nicaragua, Honduras, and Costa Rica were classified as grouping with the Northern morphotype (Table S2). Using a reduced set of landmarks, the specimen from Panama was attributed to the Central morphotype. When performing these classification methods using the four groups as factors (i.e., Northern, Central, Southern, and Guianan morphotypes; see Table S3), instead of countries, we retrieved 100% correct classification of specimens.

We performed similar analyses (PCA and LDA, see Fig. S5) using linear cranial measurements traditionally used in systematic studies. In all cases, these analyses failed to retrieve a clear-cut discrimination between the four groups defined above.

## DISCUSSION

### Morphological variation of skull among *Dasypus* species

Skull ratios are commonly used to compare *Dasypus* species, especially the length of the palate to the length of the skull (PL/CNL), and the length of the rostrum to the length of the skull (RL adj./CNL) (*Wetzel, 1985*). Three subgenera are commonly recognized on this basis: *Cryptophractus* (including *Dasypus pilosus*), *Hyperoambon* (including *Dasypus kappleri*), and *Dasypus* (including all remaining species) (*Wetzel & Mondolfi, 1979*). Our results are largely consistent with findings from previous studies regarding existing differences among *Dasypus* species. Allometry substantially explains cranial differences, with the exception of *Dasypus pilosus*, which does not follow the main dasypodine allometric trend (Figs. 3 and 4). The hairy long-nosed armadillo is clearly distinguished from the other four *Dasypus* species studied by a lengthening of the snout and mandible and a lesser development of the braincase and basicranium. All these characteristics
might be linked to their unique diet, which predominantly includes ants and termites (*Castro et al., 2015*). Considering these distinctive morphological features and the specific structure of its osteoderms, *Castro et al. (2015)* recently proposed including *Dasypus pilosus* in a different genus, *Cryptophractus*. However, recent molecular results (*Gibb et al., 2016*) did not support such a taxonomic reassessment and argued for the retention of the hairy long-nosed armadillo in the genus *Dasypus*. *Dasypus pilosus* thus likely represents a case of rapid acquisition (i.e., 2.8 Ma as estimated by *Gibb et al., 2016*) of distinctive morphological traits in line with the shift to a divergent behavior, ecology, and high altitude habitat.

Both molecular and morphological data suggested that *Dasypus kappleri* is broadly separated from the other *Dasypus* species (*Wetzel & Mondolfi, 1979*; *Gibb et al., 2016*), which contrasts with recent results of *Castro et al. (2015)* in which it is the crown group. Mitogenomic data clearly identified *Dasypus kappleri* as the sister group to all other *Dasypus* species, from which it diverged more than 12 million years ago (*Gibb et al., 2016*), and *Gibb et al. (2016)* suggested placing it in the distinct genus *Hyperoambon*, as originally proposed by *Wetzel & Mondolfi (1979)*. *Dasypus kappleri* exhibited significant morphological differentiation in our results, with all the specimens congregating in the morphospace at least in part because they are much larger than the other species. However, the cranial morphology of *Dasypus kappleri* still remains very close to that of *Dasypus novemcinctus* when compared to that of *Dasypus septemcinctus*, *Dasypus hybridus*, and *Dasypus pilosus* (Fig. 4). *Dasypus kappleri* also grouped with *Dasypus novemcinctus* to the exclusion of these latter taxa in *Castro et al. (2015)*. Recently, *Feijo & Cordeiro-Estrela (2016)* proposed recognizing three species within *Dasypus kappleri* based on morphological differences of the skull and carapace: *Dasypus kappleri* distributed in the Guiana shield; *Dasypus pastasae* occurring from the eastern Andes of Peru, Ecuador, Colombia, and Venezuela south of the Orinoco River into the western Brazilian Amazon; and finally *Dasypus beniensis* that occurs in the lowlands of the Amazonian Brazil and Bolivia to the south of the Madre de Dios, Madeira, and the lower Amazon rivers. We only had access to a limited number of specimens, but did not retrieve such a clear geographical segregation in shape (Fig. S6A), while we observed a mild differentiation in size with the Guianan *Dasypus kappleri* usually being larger (Fig. S6B).

*Wetzel & Mondolfi (1979*: 47*)* placed *Dasypus septemcinctus*, *Dasypus hybridus*, and *Dasypus sabanicola* in the same subgenus together with *Dasypus novemcinctus*. We observed that *Dasypus hybridus* and *Dasypus septemcinctus* group together in the morphospace, but are largely separated from *Dasypus novemcinctus*. These two species are usually distinguished by external features, *Dasypus hybridus* possessing shorter ears and a longer tail than *Dasypus septemcinctus* (*Hamlett, 1939*; *Wetzel & Mondolfi, 1979*). They were considered as valid based on cranial and body measurements (*Hamlett, 1939*; *Wetzel, 1985*) despite the fact that they display many external resemblances and have overlapping geographical distributions. Our morphometrical results showed that *Dasypus hybridus* and *Dasypus septemcinctus* display very similar cranial and mandibular morphologies. They also display several cranial characteristics in common with juvenile specimens of *Dasypus novemcinctus*. Such morphological similarities echo recent molecular findings

(*Gibb et al., 2016*) which showed that mitogenomic sequences of *Dasypus hybridus* were almost identical to those of an Argentinian *Dasypus septemcinctus* (99.3% identity). A recent study of their internal cranial sinuses also failed to provide diagnostic characters for distinguishing these two species (*Billet et al., 2017*). Our samples were very limited for both *Dasypus hybridus* (*n* = 4) and *Dasypus septemcinctus* (*n* = 3), but additional sampling will undoubtedly help to define the systematic status of the two species.

We did not have access to the two most recently recognized *Dasypus* species: the Yunga's lesser long-nosed armadillo *Dasypus mazzai* (*Yepes, 1933*; *Vizcaíno, 1995*; *Feijo & Cordeiro-Estrela, 2014*), and the northern long-nosed armadillo *Dasypus sabanicola* (*Mondolfi, 1967*). The validity of the former was and is still hotly debated (*Wetzel & Mondolfi, 1979*; *Vizcaíno, 1995*; *Gardner, 2008*; *Feijo & Cordeiro-Estrela, 2014*), while the specific status of the latter also remains controversial (*Wetzel & Mondolfi, 1979*; *Wetzel, 1985*). Cranial morphometric data might provide useful insights into the systematic status of the two species.

## Morphological systematics and skull shape variation in *Dasypus novemcinctus*

Relative skull shape has previously been examined for systematic purposes in the genus *Dasypus*, but never with a focus on patterns of intraspecific variation. *Hamlett (1939)* cast doubt on the possibility of reliably identifying different subgroups within *Dasypus novemcinctus*, although early workers had already recognized several, either at a specific or at a subspecific level. *Peters (1864)* described *Dasypus fenestratus* from Costa Rica based on the position of the small and numerous major palatine foramina, some of which are connected to the incisive foramina through a groove between (not in front of) the anterior teeth, as well as its medially shorter maxillo-palatine bones, the position of the palatine suture posterior to the end of the tooth row, the position of the lacrimal foramen closer to the orbital rim, and one character related to the extent of the pelvic shield of the carapace. *Gray (1873)* tentatively recognized as many as seven species of nine-banded armadillos in South and Central Americas, among which five were new: *Tatusia* (=*Dasypus*) *granadiana*, *Tatusia leptorhynchus*, *Tatusia brevirostris*, *Tatusia leptocephala*, and *Tatusia boliviensis*. He also followed *Peters (1864)* and recognized *Tatusia mexicana* (a variety of *Dasypus novemcinctus* in *Peters, 1864*), but decided to ignore *Tatusia fenestratus*. Both *Peters (1864)* and *Gray (1873)* used a very small number of specimens, and *Gray (1873)* distinguished all these species based mainly on the morphology of the lacrimal bones and minute morphological variations of the head scutes. *Allen (1911)* later considered *Dasypus fenestratus* and *Dasypus mexicanus* as synonyms at the subspecific level (*Dasypus novemcinctus fenestratus* Peters). He also described *Dasypus novemcinctus hoplites* from Grenada, a subspecies that he considered to be distinctly characterized by a shorter tooth row, due to the absence of the last tooth locus.

From the inspection of a series of specimens from Panama, Costa Rica, and Yucatan, *Allen (1911)* also distinguished a Central American morphotype. Compared to Brazilian specimens, Allen's Central American armadillo was characterized by short palatine bones that did not reach the level of the most posterior teeth, an obvious inflation of the maxillary region located in front of the lacrimal bone, and a lateral margin of the skull

that was largely convex at the level of the second or third tooth in ventral view. Based on size differences, *Hagmann (1908)* described the subspecies *Dasypus novemcinctus mexianae*, which he thought was restricted to a small area close to the mouth of the Amazon River. *Lönnberg (1913)* defined *Dasypus novemcinctus aequatorialis* from Ecuador, which *McBee & Baker (1982)* later proposed considering a probable synonym to *Tatusia granadiana Gray (1873)*. Lönnberg's comparisons were based on morphological characteristics of the carapace, *Dasypus novemcinctus aequatorialis* showing differences in the occipital portion of the frontal shield, as well as different proportions of the scales of the shoulder and pelvic shields. Later on, *Russell (1953)* proposed recognizing two subspecies in Mexico: *Dasypus novemcinctus davisi* in the northwestern part of Mexico, and *Dasypus novemcinctus mexicanus* in the remaining part of the country. Even if it is close morphologically to *Dasypus novemcinctus mexicanus*, *Dasypus novemcinctus davisi* is much smaller in size, and displays a few distinctive features, such as small maxillary teeth, a narrow mandible with a posteriorly projected angular process, and differences in suture closure patterns and shape, for instance with a parietal-frontal sutures that lies well behind the posterior process of the zygomatic arch (*Russell, 1953*). Most of these early descriptions, be they at a specific or subspecific level, were based on subtle morphological differences. No proper quantification of the skull variation was undertaken up until now.

Our statistical analysis of skull shape demonstrated that *Dasypus novemcinctus* exhibits a significant level of intraspecific variation, with several clearly identified subgroups. Our multivariate analyses suggest the absence of sexual dimorphism in the shape of the cranium and a slight sexual dimorphism in the mandibular morphology. *McBee & Baker (1982)* proposed that male nine-banded armadillos tend to be slightly larger than females. However, *Loughry & McDonough (2013)* pointed out that McBee and Baker based this assertion on a paper that only examined a very small number of animals, and reach the more general conclusion that there is no sexual dimorphism in this species using measurements of hundreds of specimens. Our univariate analyses confirmed the absence of size differences between females and males *Dasypus novemcinctus*. We show that allometry is likely to explain a substantial part of the observed morphological variation, including the geographical variation. Our morphometric analysis revealed distinct geographical differentiation among nine-banded armadillos, both in size and shape. This echoes early findings by *Wetzel & Mondolfi (1979)* who already pointed out size gradients among different populations of *Dasypus novemcinctus*. Interestingly, our geometric morphometric analyses permitted us to define four discrete phenotypic units. These units display very different cranial characters and occupy very distinct geographical distributions, which are in essence allopatric.

Specimens from Brazil, Uruguay, Paraguay, Bolivia, Peru, and from regions of Ecuador, Colombia, and Venezuela located east of the Andes make up most of one group and show a very stable pattern of variation (Fig. 9). They are on average smaller than the three remaining groups. *Loughry & McDonough (1998)* already showed that body size of armadillos from Poco das Antas (Brazil) was smaller than that of animals from the United States. Skulls of this Southern morphotype are characterized by smaller and flatter skulls with short frontal sinuses, a narrow snout with short premaxillary bones, a narrow

interorbital width, a long and slender jugal part of the zygomatic arch, long pterygoid processes, and a basicranium aligned with the palate in lateral view (Fig. 6). We found no sign of morphological differentiation of specimens from the mouth of the Amazon River, despite of the recognition of the subspecies *Dasypus novemcinctus mexianae* (*Hagmann, 1908*). The area covered by the specimens attributed to this morphological unit fully encompasses the Amazon basin and seemed to be delimitated by the Andes on the western side. As a matter of fact, the single Ecuadorian specimen coming from the eastern side of the Andes appeared to be distinct from most other Ecuadorian specimens, but morphologically close to Brazilian and Bolivian specimens. The same holds true for the Peruvian, Colombian, and Venezuelan specimens collected east of the Andes. The distribution of this group is reminiscent of that of the subspecies *Dasypus novemcinctus novemcinctus* Linnaeus, except for the Guiana Shield area (*Gardner, 2008*). It also recalls a similar lineage molecularly identified (M.-C. Arteaga, 2017, unpublished data) and the Southern morphotype evidenced by the analysis of paranasal spaces (*Billet et al., 2017*). Unfortunately the type specimen of *Dasypus novemcinctus*, which is supposedly housed in the Swedish Museum of Natural History in Stockholm (*Lönnberg, 1913*), could not be included in our analyses. The type locality of *Dasypus novemcinctus* Linnaeus is "America meridionali" and is generally thought to be from the eastern coast of Brazil (*Allen, 1911*).

The next differentiated group is represented by individuals originating from the Guiana shield region, including French Guiana, Guyana, Suriname, and Amapa in Brazil (Fig. 9). All the specimens belonging to this Guianan morphotype display large dome-shaped skulls that share distinctive morphological features, including long frontal sinuses, a wide snout with long premaxillary bones, a large interorbital width, large lacrimal bones, a short and massive jugal part of the zygomatic arch, an anterior border of the palatine located well behind the posterior end of the tooth row, short pterygoid processes, and a basicranium situated above the palatal plane (Fig. 6). Studies of paranasal sinuses affirm the distinctness of this group and show that the dome-shaped frontal region of Guianan nine-banded armadillos is occupied by a characteristically inflated pair of frontal sinuses that extend posteriorly to the fronto-parietal suture (*Billet et al., 2017*). No subspecies has ever been recognized or proposed in this part of South America, and such a clear-cut morphological divergence of Guianan specimens of *Dasypus novemcinctus* is here proposed for the first time. These morphometric findings corroborate recent molecular studies, which showed that specimens from French Guiana are very distant from the US populations (*Huchon et al., 1999*) and represent a distinct branch in the dasypodine mitogenomic tree (*Gibb et al., 2016*).

The distribution of the third recognized morphological group is more limited. It is distributed from the western Andes of Ecuador and Colombia to Venezuela, and Panama (Fig. 9). This Central morphotype is characterized by high and short skulls having moderately developed frontal sinuses, long premaxillary bones, a narrow interorbital width (wider than Southern specimens but narrower than Guianan specimens), a massive anterior part of the zygomatic arch that is much larger than the posterior part, a short and high jugal part of the zygomatic arch that is largely convex ventrally, an anterior border of the palatine located well behind the posterior end of the tooth row, short

pterygoid processes, and a basicranium well above the palatal plane (Fig. 6). This distribution roughly corresponds to the combined ranges of two previously described subspecies, *Dasypus novemcinctus fenestratus* (*Peters, 1864*) and *Dasypus novemcinctus aequatorialis* (*Lönnberg, 1913*) suggesting that these subspecies might be synonymous. However, we could not fully test this hypothesis since we had access to only one specimen from west of the Andes in Peru, Ecuador, and southern Bolivia. Studies on the paranasal spaces (*Billet et al., 2017*) failed to recognize such a group, and instead grouped some specimens from these regions with specimens from North and Central America, whereas others (from the western parts of Colombia, Venezuela and from Panama) could not confidently be referred to a given frontal sinus morphotype. In contrast, molecular studies recovered a lineage similar to the group recognized here, distributed from the Northern Andes to Central America and then expanding into western Mexico (*Arteaga et al., 2012*).

The last distinct morphotype occurs from Nicaragua to the southeastern part of the USA (Fig. 9). The range of this Northern morphotype spans the proposed distribution areas of the subspecies *Dasypus novemcinctus mexicanus* and *Dasypus novemcinctus davisi*, as well as the northernmost part of the distribution range of *Dasypus novemcinctus fenestratus*. All the skulls from this area display moderately developed frontal sinuses convergent toward the midline, long premaxillary bones, a narrow interorbital width (wider than Southern specimens but narrower than Guianan specimens), a long and slender jugal part of the zygomatic arch that is largely convex ventrally, an anterior border of the palatine located at the level of the posterior end of the tooth row, short pterygoid processes, and a basicranium slightly above the palatal plane (Fig. 6). Contrary to *Russell (1953)*, we did not find major morphological cranial differences between north-western and eastern Mexican populations. Our results thus cast doubts on the validity of the subspecies *Dasypus novemcinctus davisi*. The morphological homogeneity in this group is also at odds with the presence of two mitochondrial lineages in Mexico (*Arteaga et al., 2012*), but is consistent with the presence of nuclear gene flow between them (*Arteaga et al., 2011*). This Northern group includes the invasive US armadillo population, which is derived from two geographical sources: one from Mexico, and one from south-central Florida, where captive animals were presumably released (*Loughry & McDonough, 2013*). Echoing the results obtained on six microsatellite loci described by *Loughry et al. (2009)*, our findings indicate a close relationship between the two US populations. The recognition of this Northern unit with individuals ranging from Central to the southeastern USA is also in agreement with their distinctive pattern of paranasal sinuses (*Billet et al., 2017*).

The newly recognized subgroups within *Dasypus novemcinctus* prompt questions about the role of ecological factors likely to have influenced their morphological differentiation. Variation in skull morphology as a result of ecological factors has been studied in a number of species over recent years (*Caumul & Polly, 2005*; *Wroe & Milne, 2007*; *Hautier, Lebrun & Cox, 2012*). Factors such as temperature, diet and competition may cause phenotypic variation and are likely to explain some morphological differences among the identified groups. These ecological factors vary in relation to geography, and differences in geographical distribution can drive selection for different phenotypes, which may

eventually lead to distinctive populations or even new species. Since the four *Dasypus novemcinctus* subgroups are not sympatric in most of their respective natural ranges, we can hypothesize that environment and/or genetic drift, but not competition, may be responsible for some of the observed intraspecific variation. The northern Andes constitute a clear geographical barrier, which limited contacts between northern/central and southern populations, and thus has likely played a major role in shaping the morphological differentiation of the long-nosed armadillos. This biogeographical barrier seems to have played a significant role in xenarthran evolution and diversification in general (*Moraes-Barros & Arteaga, 2015*), since it also marks the separation between the two living species of tamanduas, with *Tamandua mexicana* in the north and *Tamandua tetradactyla* in the south (*Superina, Miranda & Abba, 2010*), as well as between two living species of within naked-tailed armadillos, with *Cabassous centralis* in the north and *Cabassous unicinctus* in the south (*Abba & Superina, 2010*).

The geographical distribution of the divergent populations of *Dasypus novemcinctus* recalls the pattern of morphological differentiation recently proposed for the greater long-nosed armadillo (*Dasypus kappleri*), especially for the Guianan specimens (*Feijo & Cordeiro-Estrela, 2016*). However, in the nine-banded armadillo, we did not find a clear morphological differentiation within the Amazonian basin, as defined by the opposite banks of the Madeira-Madre de Dios rivers (*Feijo & Cordeiro-Estrela, 2016*), which separate *Dasypus pastasae* from *Dasypus beniensis*. Given the extent of morphological variation reported within *Dasypus kappleri*, *Feijo & Cordeiro-Estrela (2016)* interpreted their findings as indicative of the fact that this species complex diverged earlier than other *Dasypus* species, which would allow them to accumulate more differences. Such a hypothesis seems difficult to support, in view of the substantial morphological variation observed among different populations of *Dasypus novemcinctus*, which have diverged more recently (3.7 Ma, *Gibb et al., 2016*). *Feijo & Cordeiro-Estrela (2016)* also proposed that such cumulative differences may result from strong environmental selective pressures. The newly discovered morphological diversity within *Dasypus kappleri* and *Dasypus novemcinctus* is likely to represent parallel cases of allopatric differentiation in response to diverging environmental pressures. In both cases, only the future collecting of large-scale genomic nuclear data will allow testing of these taxonomic proposals based on morphological data.

## CONCLUSION

If our interspecific comparisons primarily just provide additional support for the current designation of each group as a separate taxon, the intraspecific analyses of variation within *Dasypus novemcinctus* enable detection of previously overlooked morphotypes. Our study of the intraspecific variation of the skull in *Dasypus novemcinctus* reveals clear links to geographic distribution, and allows a revision of past taxonomic delimitations. Based on the cranial differences observed, we consider that *Dasypus novemcinctus* should be regarded either as a polytypic species (with three to four subspecies) or as a complex of several species (with three to four species). A new unit of nine-banded armadillos from the Guiana Shield could be established, which is in agreement with most recent investigations of molecular data and internal anatomy (*Loughry & McDonough, 2013*; *Billet et al., 2017*).

At this stage, we refrain from making definitive taxonomic pronouncements for the four morphological subsets in the absence of a more exhaustive molecular sampling allowing a complete taxonomic reassessment of the genus (M.-C. Arteaga, 2017, unpublished data). The discovery of divergent populations within *Dasypus novemcinctus* has implications for conservation of the species. In some areas, human activities have led to habitat degradation and fragmentation (*Zimbres et al., 2013*) or even to habitat loss. These divergent populations may be under threat and may require specific conservation measures, or at least a close re-examination of their conservation status. If we were to consider them as separate management units and not as a single species with a large distribution, the threat of endangerment to *Dasypus novemcinctus* may merit re-evaluation, since it is currently classified globally as "Least Concern" by the IUCN (*Loughry, McDonough & Abba, 2014*). In addition, our results demonstrate that specimens of *Dasypus novemcinctus* should be chosen with caution when making anatomical comparisons or performing cladistic analyses (*Castro et al., 2015*); their geographical distribution should be at least specified in all cases. Lastly, morphological investigation of intra- and interspecific variation in *Dasypus* needs to be extended to the other parts of the body, the carapace in particular. The cranial differences detected among the defined groups might be linked to previously detected differences in the number and shape of scutes on the head shield (*Lönnberg, 1913*), for example. Geometric morphometric data holds out the possibility of effectively studying covariation patterns among osteological parts and features of the carapace. Given the quality of the cingulate fossil record, using geometric morphometric methods seems likely to be equally useful on extinct forms, and so might also provide fruitful ways to interpret past morphological diversity.

## ACKNOWLEDGEMENTS

We are grateful to Christiane Denys, Violaine Nicolas, and Géraldine Véron, (Muséum National d'Histoire Naturelle, Paris, France), Roberto Portela Miguez, Louise Tomsett, and Laura Balcells (British Museum of Natural History, London, UK), Eileen Westwig (American Museum of Natural History, New York, USA), Burton Lim (Royal Ontario Museum, Toronto, Canada), Nicole Edmison and Chris Helgen (National Museum of Natural History, Washington, DC, USA), Jake Esselstyn (Louisiana State University, Museum of Natural Sciences, Baton Rouge, USA), Manuel Ruedi (Muséum d'Histoire naturelle, Geneva, Switzerland), Claudia Regina da Silva (Instituto de Pesquisas Científicas e Tecnológicas do Estado do Amapá, Macapá, Brazil), Steven van der Mije (Naturalis Biodiversity Center, Leiden, Netherland), François Cazeflis and Suzanne Jiquel (Institut des Sciences de l'Evolution, Montpellier, France), Lucile Dudoignon (KWATA association, Cayenne, France), Pierre Charles-Dominique (CNRS, Montpellier, France), Maria-Clara Arteaga, Maria Nazareth da Silva (Instituto de Pesquisas Científicas e Tecnológicas do Estado do Amapá, Macapá, Brazil) for access to comparative material. We thank Clara Belfiore for her help in the data acquisition and Rémi Lefebvre for interesting discussions on the dataset. R. Lebrun (Institut des Sciences de l'Evolution, Montpellier, France), Farah Ahmed (British Museum of Natural History, London, UK), Miguel García-Sanz and

Florent Goussard (Platform AST-RX MNHN, Paris, France) generously provided help and advice on the acquisition of CT scans. Some of the experiments were performed using the μ-CT facilities of the Montpellier Rio Imaging (MRI) platform and of the LabEx CeMEB. In compliance with Advantages and Benefits Sharing policy in French Guiana, material from French Guiana has been registred in the collection JAGUARS (http://kwata.net/la-collection-jaguars-pour-l-etude-de-la-biodiversite.html; CITES reference: FR973A) supported by Kwata NGO, Institut Pasteur de la Guyane, DEAL Guyane, and Collectivité Territoriale de la Guyane. This is contribution ISEM 2017-128 of the Institut des Sciences de l'Evolution.

### Funding

This work has benefited from an "Investissements d'Avenir" grant managed by Agence Nationale de la Recherche, France (CEBA, ref. ANR-10-LABX-25-01). This research received support from the Synthesys Project (http://synthesys3.myspecies.info/), which is financed by the European Community Research Infrastructure Action under the FP7. The funders had no role in study design, data collection and analysis, decision to publish, or preparation of the manuscript.

### Grant Disclosures

The following grant information was disclosed by the authors:
Agence Nationale de la Recherche, France: CEBA, ref. ANR-10-LABX-25-01.

### Competing Interests

The authors declare that they have no competing interests. Benoit de Thoisy is an employee of Association Kwata, Cayenne, French Guiana.

### Author Contributions

- Lionel Hautier conceived and designed the experiments, performed the experiments, analyzed the data, contributed reagents/materials/analysis tools, wrote the paper, prepared figures and/or tables, reviewed drafts of the paper.
- Guillaume Billet conceived and designed the experiments, contributed reagents/ materials/analysis tools, reviewed drafts of the paper.
- Benoit De Thoisy contributed reagents/materials/analysis tools, reviewed drafts of the paper.
- Frédéric Delsuc conceived and designed the experiments, contributed reagents/ materials/analysis tools, reviewed drafts of the paper.

### Data Availability

The Procrustes coordinates of all specimens used in our analyses are uploaded as Supplemental Dataset Files. All the analyses discussed in the paper can be performed using these Procrustes coordinates.

## Supplemental Information

Supplemental information for this article can be found online at http://dx.doi.org/10.7717/peerj.3650#supplemental-information.

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
