# Peer review of "Beyond the carapace: skull shape variation and morphological systematics of long-nosed armadillos (genus Dasypus)"

_PeerJ, doi:10.7717/peerj.3650_

## Round 0.1 · original submission · Minor Revisions

Dear authors,

I have received comments on your manuscript from three reviewers. All reviews are highly positive about the preparation, content and importance of your study, and recommend it be published following minor revisions. I agree with their comments that this is an excellent contribution. In particular, please address the suggestion by Reviewer #1 and #3 that the manuscript be more prescriptive for the reader regarding the current (results-informed) state of Dasypus taxonomy. Reviewer #2 has listed (in an attached pdf) some concerns regarding the results of the PCA and regressions, please address these comments in your response.

I look forward to receiving your revised manuscript.

·

Basic reporting

See general comments.

Experimental design

No comment.

Validity of the findings

See general comments.

Additional comments

This paper is welcome and long overdue, as the systematics of the Dasypus genus have not been critically evaluated in some time. Typical for this group of authors, the analyses are thorough and well done. While the interspecific comparisons primarily just provide additional support for the current designation of each group as a separate taxon, the more important findings come from the intraspecific analyses of variation within D. novemcinctus.

My only major comment is that I am not sure where we are now with Dasypus taxonomy. For example, the authors note linkages between previously defined subspecies and the four morphotypes they identified within D. novemcinctus, but leave it at that. I am completely unclear as to which subspecies names should be retained, which eliminated, and if any new names should be used. Likewise, the discussion of the D. kappleri complex leaves open whether the three species proposed by Feijo and Cordeiro-Estrela (2016) are really legitimate and should be adopted. Such ambiguity is not all that helpful for anyone tasked with making conservation assessments or other overviews of this group. Perhaps the other papers associated with this project will make more definitive statements regarding these matters but, based on this paper alone, I would be unable to tell anyone exactly how many species are in the genus, nor what subspecies are found in D. novemcinctus. More specific comments, tied to line numbers, follow.

Throughout: There are a number of minor grammatical errors in English usage. These are too numerous to list here, but the ms will need to be thoroughly edited to correct them all.

41: The authors are apparently following Feijo and Cordeiro-Estrela (2014) in using D. mazzai instead of D. yepesi. However, they are not consistent in this usage (see line 113), and yepesi is still used by the IUCN. They may need to explicitly address this choice and clarify that mazzai and yepesi are synonyms.

58-60: McBee & Baker and Smith & Doughty are fairly dated sources for the distribution of D. novemcinctus. Better would be the IUCN web page for the species, or Loughry & McDonough (2013).

117-118: Criteria are given for how adult specimens were identified but nothing is said about how subadults and juveniles were separated. Only juveniles were used in the analyses (lines 226-233), so it is not clear whether the subadult designation was even used. Along these lines, the specimen information in S1 provides no age information, thus it is not clear how many adults, subadults and juveniles were measured.

167: Lumping specimens by country seems artificial as political boundaries have little biological relevance. Admittedly, the analyses wind up eliminating these boundaries in identifying morphotypes but I wonder if they are really necessary in the first place.

243-245: Is there any relationship between sex and size?

246-254: Given the very low percent of total variation explained by PC1 and PC2 in this analysis (and the low R2 value in Fig. 5), I wonder if anything meaningful can really be said about mandibular differentiation across countries.

329: “four” is a bit confusing. I finally realized the authors were talking about the four other species included in the analyses, but I initially interpreted this as referring to all other Dasypus species, of which there are more than four (presumably).

369: Should it be “two species” and not “two genera”?

424-425: Loughry & McDonough (2013) pointed out that citing McBee & Baker for the occurrence of sexual dimorphism in D. novemcinctus is problematic because they based this assertion on a paper that only examined a very small number of animals. The more general conclusion that Loughry & McDonough reach, based on measurements of hundreds of animals, is that there is no sexual dimorphism in this species.

436: Loughry & McDonough (1998, Revista de Biologia Tropical) showed that body size of armadillos from Poco das Antas, Brazil was smaller than that of animals from the U.S., which supports the findings of the current study.

527-531: Moraes-Barros & Arteaga (2015; J. Mammal.) have made similar points about the role of the Andes in the evolution of xenarthrans.

Bibliography: There are some inconsistencies in how article titles are formatted.

652-654: Species names should be italicized (see also line 741)

714: It is not clear what type of publication this is. Is it a thesis, or what? (see also line 746)

723: Should be Brazil and United, not brazil and united.

886 & 895: term “specimen” not italicized?

Figure 1: Not sure how easy it would be to do this, but it would be helpful to number the landmarks on the cranium and mandible in order to link them with the definitions in Tables 1 and 2.

Reviewer 2 ·

Basic reporting

See attached file.

Experimental design

See attached file.

Validity of the findings

See attached file.

Additional comments

Please see my all of my comments in the attached file.

Annotated reviews are not available for download in order to protect the identity of reviewers who chose to remain anonymous.

·

Basic reporting

The basic execution of this paper is excellent. There are some minor English language issues and other minor lapses (unsurprising given that none of the authors is a native speaker of English). I have marked these and suggested corrections in the attached, annotated pdf. The paper is well written and well organized. The figures are well done for the most part, but I do have a couple of suggestions for improvement. Firstly and most importantly, the authors need to add brief explanations of the mandibular and skull figures accompanying the graphs in Figures 3-8. Right now, there is no mention of the images or what they mean (though it is described in the text). Secondly, there are a lot of different symbols used in the graphs, and often their distributions are overlapping, making it difficult to visualize patterns. I think that drawing circles around groups that are currently distinguished only by color (as was done in the Wetzel 1985 paper cited in the text), i.e., around the species in Figs. 3 and 4, and around the broad geographic groupings in Figs 5-8, might make these patterns easier for the reader to discern. This might also be helpful in reading some of the graphs from the supplemental materials.

Experimental design

Once again, the authors have done an excellent job with this aspect of their paper. The goals and justification of the research are explained clearly, as is the methodology. This is a significant piece of work on an important, widespread genus of Neotropical mammals (the nine-banded armadillo genus Dasypus), one with significant interactions with humans, and which interestingly has both expanding and endangered populations, and it tackles real unresolved issues in the basic taxonomy of the genus using novel (and highly appropriate) imaging, analytical and statistical methods. The only real issues I have with the experimental design have to do with sample size. I am a little troubled by the low sample sizes for some of the more poorly known, poorly studied species (4 specimens for D. hybridus, 3 for D. septemcinctus, and 2 for D. pilosus), and am very troubled that two living species are missing entirely from the analysis (D. sabanicola and D. mazzai). I believe that this detracts from the power of the analysis that the authors conduct, makes their case less compelling than it otherwise could be, and perhaps leaves them hesitant to make definitive taxonomic pronouncements. Of course, the situation is understandable. Specimens of these lesser-known, less abundant, and therefore poorly studied species (often restricted to fairly remote regions of SouthAmerica) are hard to come by, and so I don't necessarily fault the authors. Nor am I suggesting that this should be a barrier to the publication of this study. However, even a small increase in sample size would make this a stronger study, I think. I might encourage the authors to avail themselves of resources like VertNet (portal.vertnet.org), which allows one to search the content of a large number of significant museum collections of mammals worldwide. My own quick search revealed 5 specimens of D. sabanicola (2 at the Univ. of Florida, 3 at the Univ. of Washington), as well as additional specimens of D. septemcinctus, D. hybrids, and D. pilosus. I understand that arranging access can be difficult and expensive, if not impossible, and so again, I don't think the authors should be required to expand their sample for this publication. However, I do offer this as a friendly suggestion.

Validity of the findings

Here too the authors have done a fine job. The data and statistics are robust, and the conclusions are explained clearly, justified by the evidence presented, and put in appropriate context. The only real issue I have is that the authors appear to be reluctant to make definitive taxonomic pronouncements. At the conclusion of this study, they have data from cranial morphometrics, and they compare these to results from a related but distinct dataset based on paranasal sinus anatomy, and to genetic data. It seems to me that they have enough information to make some new taxonomic assignments, at least at the subspecies, if not the species or genus level. For example, they discuss (lines 466-470) the distinctiveness of the Guianan populations of D. novemcinctus based on all three criteria (skull morphometry, paranasal sinus anatomy and genetics), and yet do not make a formal assignment of this group even to its own subspecies. I think that the overall trend in examining cingulate phylogeny is that the group is undersplit taxonomically - i.e., that there are more family, generic, and specific level taxa (especially if extinct taxa are included) than are currently recognized. Where there is conflicting data (e.g., with D. pilosus, where the current authors and a prior morphological phylogeny suggest the existence of a distinct genus that is not supported by molecular analysis [at least the relationships to other Dasypus are not supported - D. pilosus is still monophyletic, and could still be recognized as a distinct, if highly nested, genus within Dasypus, though it would make Dasypus itself paraphyletic]; also D. kappleri, which has some data that supports its recognition as a distinct genus, and other data that contradicts such aconclusion), the decisions to create new taxa are more difficult to justify, but I am not a believer in waiting for the next study to come along with better data before making these taxonomic decisions. In my experience, that more complete study often does not come at all, or if it does come, years or even decades may pass in the interim. And I would point out, if that latter study disagrees with the taxonomic assessments of the current authors, it can suggest new changes to the taxonomy. Taxonomy is not meant to be static, but is an important reflection of our knowledge of the biodiversity of the organisms involved, and as the authors themselves note, can even have significant conservation implications. So I would encourage the authors to consider erecting some formal taxonomic divisions based on their results.

Additional comments

Overall, the authors are to be commended for an well-executed piece of research.

---

## Round 0.2 · accepted · Accept

Dear authors,

Thank you for addressing all the comments raised previously by the reviewers. On a final review of your ms, I noted a couple of minor (editorial) points:

ln173: replace "analyzed" with "visualized" or similar (PCA is just an ordination technique)

ln180: I would rephrase this slightly, as "first Principal Components" is not informative/needed. I would simply state that PCs accounting for 90% of the interspecific shape variance in the sample were retained for analysis.

ln373: replace "bigger" with "larger"
ln590: "refrain to make" should be "refrain from making"